# Surface-confined alternating copolymerization with molecular precision by stoichiometric control

Lingbo Xing[1,3], Jie Li[2,3], Yuchen Bai [1,3], Yuxuan Lin[1], Lianghong Xiao[1], Changlin Li[1], Dahui Zhao [1], Yongfeng Wang [2] ✉, Qiwei Chen[1] ✉, Jing Liu [1] ✉ & Kai Wu [1] ✉

Keen desires for artificial mimicry of biological polymers and property improvement of synthesized ones have triggered intensive explorations for sequence-controlled copolymerization. However, conventional synthesis faces great challenges to achieve this goal due to the strict requirements on reaction kinetics of comonomer pairs and tedious synthetic processes. Here, sequence-controlled alternating copolymerization with molecular precision is realized on surface. The stoichiometric control serves as a thermodynamic strategy to steer the polymerization selectivity, which enables the selective alternating organometallic copolymerization via intermolecular metalation of 4,4″-dibromo-p-terphenyl (P-Br) and 2,5-diethynyl-1,4-bis(phenylethynyl)benzene (A-H) with Ag adatoms on Ag(111) at P-Br: A-H = 2, as verified by scanning tunneling microscopy and density functional theory studies. In contrast, homopolymerization yield increases as the stoichiometric ratio deviates from 2. The microscopic characterizations rationalize the mechanism, providing a delicate explanation of the stoichiometry-dependent polymerization. These findings pave a way to actualizing an efficient sequence control of copolymerization by surface chemistry.

Precise control of building block sequence in copolymerization is a key issue in polymer science. On the one hand, the artificial realization of sequence-controlled polymerization, which is a coding mimicry of natural DNAs, RNAs and proteins, would greatly contribute to our understanding of the delicate laws in nature. On the other hand, sequence regulation of copolymers also provides an efficient approach to tuning their physical and mechanical properties[1–3]. Among various copolymers with well-defined sequences, the alternating one manifests the simplest periodic structure, and hence is frequently used as a model system to explore the synthetic strategies aimed at sequence-controlled polymerization[4–6]. The realization of alternating copolymerization requires a higher selectivity toward cross-coupling than homo-couplings of different monomers. Such a selectivity preference

is usually realized by kinetic strategies in conventional synthesis, that is, to adopt comonomer pairs that feature a higher reaction rate for copolymerization than homopolymerization[7]. Otherwise, fastidious synthetic processes such as the iterative single unit addition method[3,8] and external fields[5,9] would be necessary to achieve the precise sequence control. Due to the strict requirements on the reaction kinetics and complicated experimental processes, efficient alternating copolymerization is extremely challenging, and hence novel strategies for precise sequence control of copolymerization are highly desirable.

The emerging on-surface chemistry, a controlled bottom-up approach to fabricating low dimensional molecular nanostructures with atomic precision[10–14], provides a potential solution to this issue. The unique solvent-free reaction condition plus two-dimensional

[1]BNLMS, College of Chemistry and Molecular Engineering, Peking University, Beijing 100871, China. [2]Center for Carbon-based Electronics and Key Laboratory for the Physics and Chemistry of Nanodevices, School of Electronics, Peking University, Beijing 100871, China. [3]These authors contributed equally: Lingbo Xing, Jie Li, Yuchen Bai. ✉e-mail: yongfengwang@pku.edu.cn; chenqw@pku.edu.cn; jing.liu@pku.edu.cn; kaiwu@pku.edu.cn

confinement and template effects of the substrate surfaces have shown their efficient controls of surface-confined reactivity and selectivity in a variety of systems[15–21]. These successes inspire the idea to explore the realization of sequence-controlled copolymerization via on-surface chemistry, which is yet to be addressed, to the best of our knowledge.

In this work, alternating organometallic copolymerization between an aromatic halide and a terminal alkyne achieved via their metalation reactions with the Ag adatoms on Ag(111) is demonstrated at the molecular level by scanning tunneling microscopy (STM) measurements and density functional theory (DFT) calculations. In contrast to conventional polymer synthesis where the sequence control usually relies on the reaction kinetics, the surface-confined alternating copolymerization in this case mainly lies in the thermodynamics. Stoichiometric ratio between two monomers is introduced as an efficient parameter to steer the surface polymerization selectivity in order to achieve the selective alternating copolymerization. The stoichiometric control is realized by either the stoichiometric dosage of two monomers or the post-addition of one monomer to the samples pre-covered by the homopolymers composed of the other one. These results not only open up a route toward sequence control of copolymerization via surface chemistry but provide molecular insights into the mechanism of the selective copolymerization as well.

## Results

Both aromatic halides and terminal alkynes are frequently used monomers in on-surface synthesis, and their surface reactivities have been well documented[22–32]. Taking the case on Ag substrates as an example, these two monomers are able to form the phenyl-silver-phenyl (P-Ag-P) and alkynyl-silver-alkynyl (A-Ag-A) intermolecular organometallic connections via the dehalogenated [Reaction (i) in Fig. 1] and dehydrogenated [Reaction (iii)] metalation reactions with the Ag adatoms under proper conditions, respectively[23,31]. When halogen substituents and alkynyl groups co-exist in a reaction system, the metalation reaction between different functional groups [Reaction (ii) in Fig. 1] also becomes possible, leading to the alkynyl-silver-phenyl (A-Ag-P) products[21,33–36]. The homo- and hetero-intermolecular organometallic connections bring the same and different building blocks together, respectively. This enables both homopolymerization (the products being denoted as $[P-Ag]_n$ and $[A-Ag]_n$, respectively) and copolymerization (the product being denoted as $[A-Ag-P-Ag]_n$) in the bi-functional system (Fig. 1). Their relative selectivity then dictates the sequence structure of the final polymeric product.

The aromatic halide and terminal alkyne monomers used in this work are 4,4″-dibromo-p-terphenyl (chemical structure shown in Fig. 2a and denoted as P-Br hereafter) and 2,5-diethynyl-1,4-bis(phenylethynyl)benzene (chemical structure shown in Fig. 2a, denoted as A-H hereafter), respectively. STM images in Fig. 2b and c depict their co-assembled structures formed on Ag(111) which co-exist with their separated domains (Supplementary Fig. 1). The A-H monomers, as

marked by the dark blue frames in Fig. 2b and c, possess a crisscross backbone, and the P-Br molecules, as marked by the green frames, look rod-like. It'll be demonstrated later on that the alternating organometallic copolymerization between these two monomers can be achieved with molecular precision by controlling the P-Br to A-H ratio (denoted as $r$). The $r$ value can be controlled by two strategies, i.e., stoichiometric dosage of two monomers at the initial stage, and post-addition of one monomer to the samples pre-covered by the homopolymers formed by the other one.

### Alternating copolymerization by stoichiometric dosage of monomers

The attempt aiming at selective alternating copolymerization began with the reaction between the P-Br and A-H monomers at a ratio of $r \sim 1$ (details for $r$ determination are provided in the Supplementary Discussion) on Ag(111) in terms of the equivalent stoichiometry of two building blocks in the target product. To initiate the surface-confined polymerization, the sample pre-covered by two monomers was annealed at room temperature (RT). However, the outcome was against our expectation: The $[A-Ag]_n$ homopolymer rather than the alternating copolymer was selectively obtained. The $[A-Ag]_n$ structures appear as the zipper-like molecular chains that are shaded in dark blue in Fig. 3b. Figure 3c is the magnified STM image of a zipper-like chain, showing the alternate arrangement of the crisscross molecular moieties and the round protrusions, which are attributed to the dehydrogenated A-H moieties and Ag adatoms, respectively. Both appearance and intermolecular separation ($1.22 \pm 0.02$ nm, as marked by the white arrow in Fig. 3c) of the zipper-like chains agree well with those of the $[A-Ag]_n$ structures formed by A-H and Ag adatoms (with an intermolecular separation of 1.21 nm) reported in our previous work[31], which supports our identification. The chemical structure of the $[A-Ag]_n$ homopolymer is illustrated in the bottom panel of Fig. 3c. At $r < 1$ (Fig. 3a), the main product is still the $[A-Ag]_n$ structures but with reduced chain lengths in comparison with the long $[A-Ag]_n$ chains obtained at $r \sim 1$.

Surprisingly, the selective formation of the $[A-Ag-P-Ag]_n$ alternating copolymer was achieved at $r \sim 2$. A representative STM image of the surface products after thermal treatment at RT is provided in Fig. 3e, displaying a high yield of the polymeric product shaded in light blue. A high-resolution STM image of the main product is given in Fig. 3f, showing the parallel alignment of the molecular chains that consist of alternately arranged crisscross and rod-like molecular moieties separated by the round protrusions. Such an appearance suggests the formation of the $[A-Ag-P-Ag]_n$ alternating copolymers that involve both P-Br and A-H molecular moieties bound to the Ag adatoms. This identification is supported by the agreement of the measured separation between the adjacent crisscross and rod-like molecules ($1.42 \pm 0.02$ nm, as marked by the white arrow in Fig. 3f) with the calculated counterpart of an A-Ag-P dimer (1.41 nm). The chemical structure of the $[A-Ag-P-Ag]_n$ copolymer is schematically shown in the

**Fig. 1 | Coexistent reaction pathways and products.** Possible reaction pathways and resulted polymeric products in surface-confined systems that contain both bromine and alkynyl functional groups.

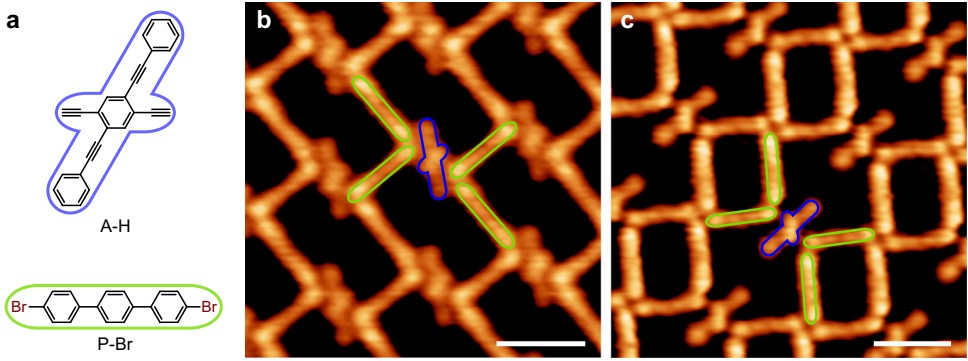

**Fig. 2 | P-Br and A-H monomers. a** Chemical structures of A-H and P-Br. **b, c** STM images of two co-assembled structures of the P-Br and A-H monomers [Scanning conditions: bias voltage $V = 10$ mV and tunneling current $I = 100$ pA for (**b**), and $V = 14$ mV and $I = 100$ pA for (**c**)]. The P-Br and A-H monomers are highlighted by the green and dark blue frames, respectively. Scale bars: 2 nm.

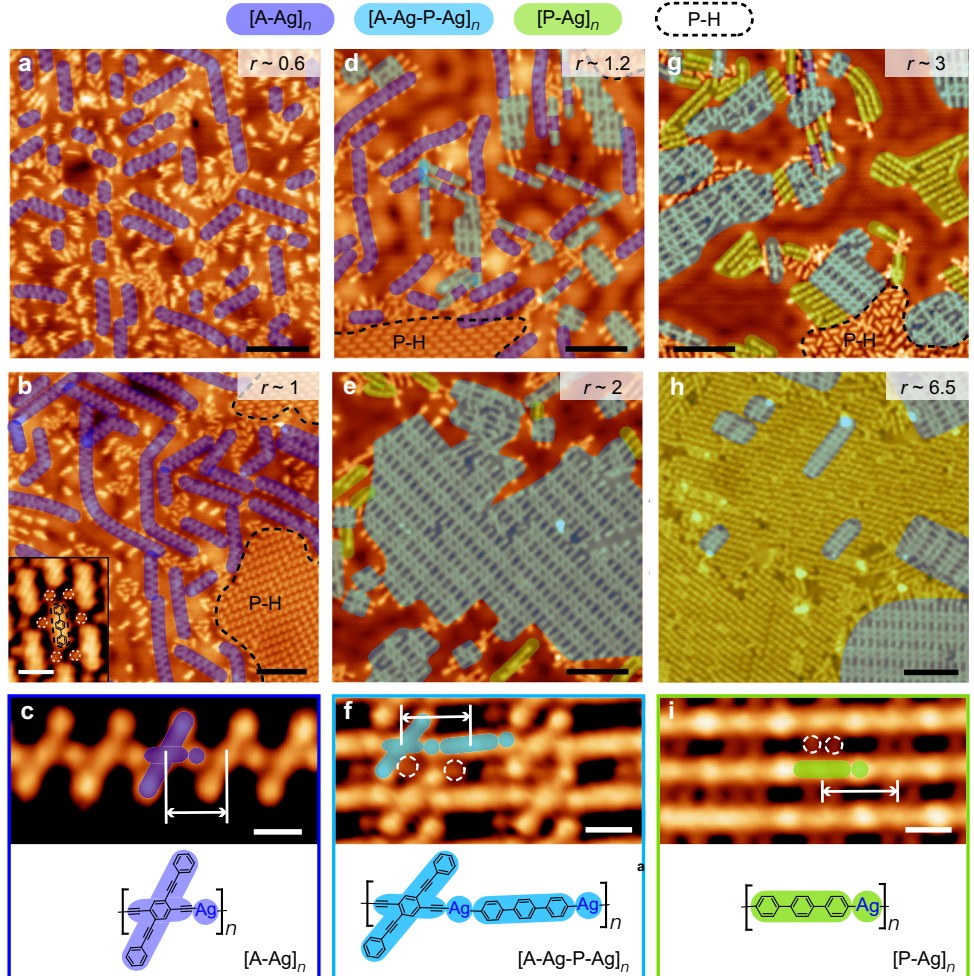

**Fig. 3 | Stoichiometry-dependent polymerization.** Large-area STM images of the reaction products achieved by P-Br and A-H on Ag(111) with (**a**) $r$ ~ 0.6 ($V = 40$ mV and $I = 100$ pA), (**b**) $r$ ~ 1 ($V = 10$ mV and $I = 100$ pA), (**d**) $r$ ~ 1.2 ($V = 10$ mV and $I = 100$ pA), (**e**) $r$ ~ 2 ($V = 100$ mV and $I = 100$ pA), (**g**) $r$ ~ 3 ($V = 100$ mV and $I = 100$ pA), and (**h**) $r$ ~ 6.5 ($V = 10$ mV and $I = 100$ pA). Scale bars: 8 nm. Inset of (**b**): High-resolution STM image of an ordered co-assembly formed by P-H and Br adatoms ($V = 500$ mV and $I = 30$ pA). Scale bar: 1 nm. High-resolution STM images of (**c**) an [A-Ag]$_n$ chain

($V = 10$ mV and $I = 100$ pA), (**f**) two [A-Ag-P-Ag]$_n$ chains ($V = 20$ mV and $I = 100$ pA), and (**i**) three [P-Ag]$_n$ chains ($V = 10$ mV and $I = 100$ pA). Scale bars: 1 nm. The white arrows mark the distances between adjacent molecular moieties in the chains. The chemical structures of the polymeric products are illustrated in bottom panels of (**c**), (**f**) and (**i**). The Br adatoms are highlighted by the white dashed circles in (**b**) inset, (**f**) and (**i**).

bottom panel of Fig. 3f. The dim dots located in between the [A-Ag-P-Ag]$_n$ chains, as marked by the white dashed circles in Fig. 3f, are the Br adatoms detached from P-Br.

A further increase in $r$ led to an augmented yield of the [P-Ag]$_n$ homopolymer. Taking the sample prepared at $r \sim 6.5$ as an example, the representative STM image (Fig. 3h) shows that the surface is dominantly covered by the parallely aligned linear polymers (shaded in green). As seen in the magnified STM image in Fig. 3i, each linear chain is composed of rod-like molecular moieties connected by round protrusions, indicative of the [P-Ag]$_n$ homopolymers formed by dehalogenated metalation of the linear aromatic halide[22,23]. The distance between the adjacent molecules, as marked by the white arrow in Fig. 3i, is measured to be $1.60 \pm 0.03$ nm. This value agrees well with previously reported periodicity of the [P-Ag]$_n$ chain formed by P-Br on Ag(111) (1.59 nm)[23], which further supports our attribution. A corresponding chemical structure is shown in the bottom panel of Fig. 3i. Detached Br adatoms appear as the dim dots located in between the [P-Ag]$_n$ chains, as marked by the white dashed circles in Fig. 3i. In addition to the main product [P-Ag]$_n$, the A-Ag-P oligomers and [A-Ag-P-Ag]$_n$ segments, as shaded in light blue in Fig. 3h, are also observed to randomly embed in the [P-Ag]$_n$ chains.

It's therefore concluded that there is a sensitive response of the polymerization selectivity to the dosage stoichiometry of these two monomers. In our experiments, when P-Br was equal to ($r \sim 1$), twice as ($r \sim 2$), and far in excess to (e.g., $r \sim 6.5$) A-H, the [A-Ag]$_n$, [A-Ag-P-Ag]$_n$ and [P-Ag]$_n$ chains turned out to be the main polymeric products, correspondingly. When $r$ fell between the specific values, a mixture of different products was observed. Specifically, both [A-Ag]$_n$ and [A-Ag-P-Ag]$_n$ products coexisted at $1 < r < 2$ (Fig. 3d), and the proportion of [P-Ag]$_n$ increased in accompany with a decrease in that of [A-Ag-P-Ag]$_n$ as $r$ grew in the regime of $r > 2$ (Fig. 3g, h). The co-existent homo- and co-polymeric segments might randomly appear in the same chain to form less ordered chain structures. It's worth mentioning that the experiments of stoichiometry-dependent reactions between A-H and P-Br were systematically carried out at varied total coverages of the monomers, verifying that the molecular coverage had a negligible effect on the polymerization selectivity. Moreover, further annealing of the A-H + P-Br samples at elevated temperatures (e.g., ~ 410 K) led to less ordered covalent structures (Supplementary Fig. 2).

## Homo- to co-polymerization transformation by monomer post-addition

In addition to controlling the dosage stoichiometry of two monomers at the initial stage of the surface-confined polymerization, the selectivity of alternating copolymerization could also be improved by the transformation of the [A-Ag]$_n$ or [P-Ag]$_n$ homopolymer induced by the post-addition of the P-Br or A-H monomer.

To address the homo- to co-polymerization transformation, the [A-Ag]$_n$ and [P-Ag]$_n$ structures were prepared in advance on Ag(111), respectively. The former was obtained by the reaction between A-H and P-Br at $r = 1$, and the latter, by debrominated metalation of pure P-Br ($r = \infty$). Subsequently, the P-Br (assembled into an ordered structure shown in the inset of Fig. 4a) and A-H (randomly adsorbed on the bare substrate, as shown in Fig. 4c and inset) monomers were added to the samples pre-covered by the [A-Ag]$_n$ and [P-Ag]$_n$ homopolymers, respectively. As such, the stoichiometric ratio between the building blocks was varied, i.e., the $r$ values being increased and decreased for the [A-Ag]$_n$ + P-Br and [P-Ag]$_n$ + A-H samples, respectively. After annealing of the samples at RT, the [A-Ag-P-Ag]$_n$ polymers or oligomers appeared in both systems, either assembled into ordered islands, as shown in Fig. 4b, or formed less ordered ensembles, as shaded in light blue in Fig. 4d. Such a homo- to co-polymerization transformation again indicates the key effect of the stoichiometric ratio between two monomers on the surface-confined polymerization selectivity.

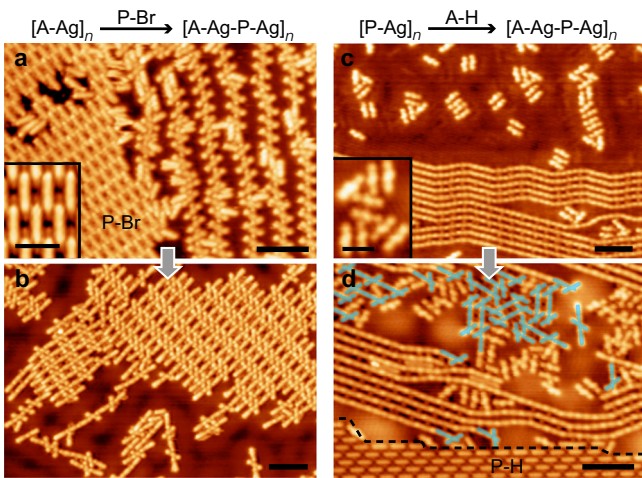

**Fig. 4 | Homo- to co-polymerization transformation.** STM images of the [A-Ag]$_n$ + P-Br sample (**a**) before ($V = 100$ mV and $I = 100$ pA), and (**b**) after being annealed at RT ($V = 100$ mV and $I = 90$ pA). Scale bars: 5 nm. Inset of (a): High-resolution STM image of the assembled P-Br molecules ($V = 20$ mV and $I = 100$ pA). Scale bar: 2 nm. STM images of the [P-Ag]$_n$ + A-H sample (**c**) before ($V = 100$ mV and $I = 100$ pA), and (**d**) after being annealed at RT ($V = 10$ mV and $I = 30$ pA). Scale bars: 5 nm. Inset of (**c**): High-resolution STM image of the A-H monomers ($V = 10$ mV and $I = 100$ pA). Scale bar: 2 nm. The A-Ag-P structures are shaded in light blue and the P-H assembly is highlighted by the black dashed frame in (**d**).

It's noticed that apart from the organometallic polymetric products, there also appeared a monomeric byproduct on the samples shown in Figs. 3 and 4. Some examples of the assembled structures containing the monomeric byproduct are highlighted by the black dashed frames in Figs. 3b, d, g and 4d. In the high-resolution STM image (inset of Fig. 3b), the monomeric product appears as a rod-like backbone, as highlighted by the black dashed frame, which is topographically similar to P-Br except for its shorter molecular length ($1.5 \pm 0.1$ nm for P-Br and $1.2 \pm 0.1$ nm for the short rod-like molecule). Such a molecular length is consistent with that of p-terphenyl (denoted as P-H, 1.2 nm)[37], a molecular analog to P-Br with two H rather than Br atoms at both ends, as shown by the chemical structure superimposed in the inset of Fig. 3b. Therefore, the short rod-like structures are assigned as the P-H molecules, which are supposed to originate from passivation of the debrominated P-Br by the H atoms that are produced from the dehydrogenation of A-H (see detailed discussion below). The passivation of the debrominated sites in P-Br is experimentally supported by the high mobility of the short rod-like molecules at 4.9 K upon tip perturbations (Supplementary Fig. 3), which indicates a weak molecule-substrate interaction. In contrast, the unpassivated species are supposed to be strongly anchored to the substrate via their debrominated sites, which would remarkably reduce their surface mobility[38,39]. The dim dots in coexistence with P-H in the assembly shown in the inset of Fig. 3b (marked by the white dashed circles) are due to the detached Br adatoms.

## Mechanism of stoichiometry-controlled alternating copolymerization

The experimental findings demonstrate that stoichiometry of different monomers plays a key role in selective alternating copolymerization. Moreover, the stoichiometric ratio that maximizes the yield of the alternating copolymer, i.e., $r = 2$, is distinct from the equivalent stoichiometry of two building blocks in the copolymeric product. To get insights into such an unexpected stoichiometric effect, the reaction mechanism of A-H and P-Br on Ag(111) including their activations and subsequent involvement in the organometallic polymerization was closely inspected.

The formation of the organometallic polymeric products clearly indicates the debrominated and dehydrogenated activations of P-Br and A-H, respectively. The debrominated activation of the aryl halides such as P-Br on metal surfaces have been well documented[25,38]. The low activation energy ($E_a$) and negative reaction energy ($E_r$, defined as the energy difference by subtracting the total energy of the reactants from that of the products) ensures the reaction to take place irreversibly at RT, giving rise to substrate-stabilized phenyl residues (denoted as P*) and detached Br adatoms on Ag(111)[25,39]. As a comparison, the activation of the alkynyl C-H bond on Ag(111) at RT becomes invalid in absence of halides according to previous reports[27,28,31]. Instead, direct C-C coupling takes place between terminal alkynes at elevated temperatures. When the alkyne monomers are modified with halogen substituents, the RT activation of the alkynyl C-H bond, however, becomes accessible to form the alkynyl-involved organometallic products[21,33,36]. A similar phenomenon was also observed in this work that the reaction of A-H in coexistence with P-Br generated the [A-Ag]$_n$ and [A-Ag-P-Ag]$_n$ products. These results clearly demonstrate the crucial role of the aryl-halide-related species in promoting the C-H bond activation in the terminal alkynes at RT.

Due to the surface reactivity of P-Br, multiple molecular species may be generated during its surface reaction, including the debrominated phenyl residue P*, organometallic structures P-Ag-P, covalent coupling product PP (denoting the polymeric one as [P]$_n$), and detached Br adatoms[23–25,38,40]. To clarify the specific P-Br-related species that takes effect on promoting the C-H bond activation in A-H, a series of control experiments with regard to the reactions of A-H with [P-Ag]$_n$ and [P]$_n$ were carried out.

The reaction between A-H and [P-Ag]$_n$ on Ag(111) at RT has already been discussed above (Fig. 4a, b), leading to the formation of [A-Ag-P-Ag]$_n$ which verifies the efficient activation of the alkynyl C-H bond in A-H. To address the reaction between A-H and [P]$_n$, the A-H monomers were evaporated onto the Ag(111) substrate pre-covered by the [P]$_n$ chains (Fig. 5a). The latter was obtained by the Ullmann coupling of P-Br at ~ 470 K[23]. The [P]$_n$ chains are featured by the homogeneous contrast along the backbone (inset of Fig. 5a), suggesting the covalent connection between the molecular building blocks. The dim dots between the [P]$_n$ chains, as highlighted by the white dashed circles in the inset of Fig. 5a, are due to the detached Br adatoms. No prominent difference was observed upon thermal treatment of the sample at RT. As shown in Fig. 5b, most A-H molecules remain intact as monomers, and the yield of the A-H-involved organometallic species, with one example shaded in dark blue in the inset, falls below 3%. Only upon thermal treatment at 330-350 K could the covalent C-C homo-coupling of A-H be initiated (Supplementary Fig. 4). The proportion of the alkynyl groups in A-H that take part in the intermolecular organometallic reactions at such a reaction temperature is ~ 8%, which is similar to that obtained by the reaction of A-H on Ag(111) at 350 K in the absence of P-Br (9%)[31], but is remarkably lower than that achieved by the reaction of A-H with P-Br or [P-Ag]$_n$ at RT.

The above-described results provide direct evidence for the distinct reactivities between A-H and different P-Br-related species. Both P-Br monomers and [P-Ag]$_n$ could effectively activate A-H on Ag(111) at RT, yielding alkynyl-involved organometallic products such as [A-Ag]$_n$ and [A-Ag-P-Ag]$_n$. In contrast, neither the [P]$_n$ species nor the Br adatom (in coexistence with [P]$_n$ and [P-Ag]$_n$) has an effect on the conversion of A-H around RT. Therefore, it's rational to propose that P* be the effective species promoting the C-H bond activation of A-H at RT, because P* is available in both reaction systems that comprise the P-Br monomers (via debromination[23,41]) and [P-Ag]$_n$ chains [via dissociation of the organometallic species[39], as experimentally confirmed by the observation of P* at the early reaction stage of the [P-Ag]$_n$ + A-H sample (Supplementary Fig. 5)], but is absent on the sample pre-covered by [P]$_n$. A possible reaction pathway for P* and the terminal alkyne is theoretically explored by DFT calculations using the

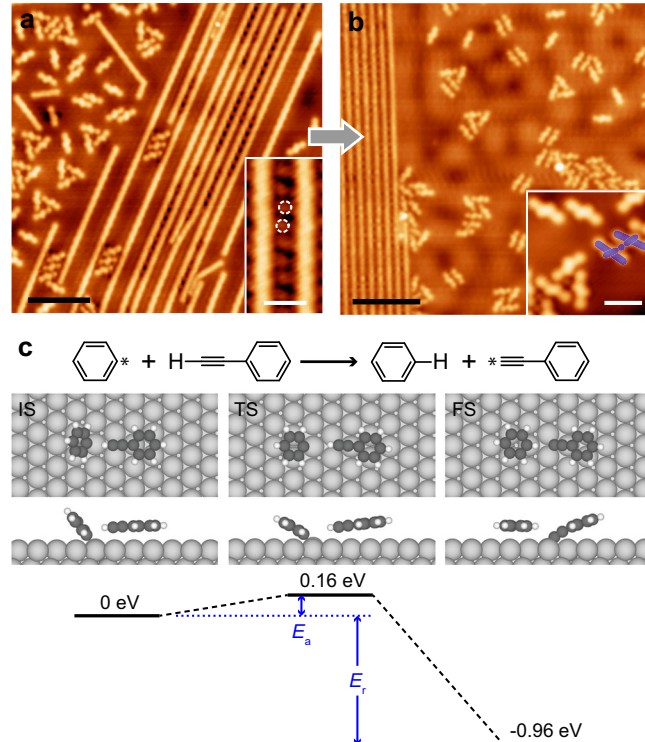

**Fig. 5 | Activation of A-H.** STM images of the [P]$_n$ + A-H sample (**a**) before ($V = 10$ mV and $I = 110$ pA), and (**b**) after being annealed at RT ($V = 100$ mV and $I = 100$ pA). Scale bars: (**a**) 6 nm, (**b**) 8 nm. Inset of (**a**): High resolution STM image of the [P]$_n$ chains ($V = 10$ mV and $I = 800$ pA). The detached Br adatoms are marked by the white dashed circles. Scale bar: 1 nm. Inset of (**b**): High resolution STM image of the unreacted A-H monomers and a A-Ag-A dimer as shaded in dark blue ($V = 10$ mV and $I = 100$ pA). Scale bar: 2 nm. (**c**) Calculated energies and molecular models of the initial (IS), transition (TS) and final (FS) states of the reaction between debrominated phenyl residue and phenylacetylene. $E_a$ and $E_r$ refer to the activation and reaction energies, respectively. Color code: dark gray for C, white for H and light gray for substrate Ag.

simplified molecular models. The proposed mechanism is provided in Fig. 5c, showing a hydrogen transfer process with an activation energy of $E_a = 0.16$ eV and a reaction energy of $E_r = -0.96$ eV. The $E_a$ remarkably lower than that of the situation without P* (~ 1.8 eV)[27,29,42] profoundly indicates the crucial role of P* in lowering the reaction energy barrier for the dehydrogenation of the terminal alkyne. Both low energy barrier and negative reaction energy of the hydrogen transfer process ensure the irreversible C-H bond dissociation in A-H at RT. Meanwhile, the P* moiety is passivated by the H atom detached from the terminal alkyne, which is consistent with the experimental observation that the P-H byproduct appears in accompany with the alkynyl-involved organometallic products like [A-Ag]$_n$ and [A-Ag-P-Ag]$_n$. Such an irreversible consumption of the P* residues would promote their formation by shifting the [P-Ag]$_n$ dissociation equilibrium in the [P-Ag]$_n$ + A-H reaction system.

As a conclusion, the activation of A-H on Ag(111) consumes P* of equivalent amount by turning the reactive species into the inert P-H molecules. Such an irreversible passivation of P* ($E_a = 1.12$ eV for the reverse reaction, Fig. 5c) also prevents P* from being involved into the reversible organometallic polymerizations that generate the [P-Ag]$_n$ or [A-Ag-P-Ag]$_n$ polymeric products. This explains the selective formation of [A-Ag]$_n$ when the amount of A-H is in excess or equivalent to that of P* generated by P-Br or [P-Ag]$_n$, i.e., at $r \le 1$, as observed in the experiments (Fig. 3a, b).

At $r > 1$, both alkynyl residue (denoted as A*) and unpassivated P* in addition to those consumed by the A-H activation would be available

in the surface-confined reaction system. These reactive molecular species can be either resulted from direct dehydrogenated and debrominated activations of the monomers or released from the organometallic polymers via the bilateral conversions of their C-Ag bonds under the reaction conditions, as reported in previous studies[43–45]. The reversibility of the organometallic reactions indicates that they are thermodynamically rather than kinetically controlled. Therefore, the maximization of the energy gain in the surface-confined reaction systems comprising both A* and P* species, once the reaction equilibrium is established, should serve as the driving force for the specific polymerization selectivity[46].

In this context, the reaction energies of homopolymerization ($E_r^{homo}$) and copolymerization ($E_r^{co}$) between A* and P* (Fig. 6a) are compared. $E_r^{homo}$ is estimated by summation of the formation energies for one P-Ag-P node and one A-Ag-A node, and $E_r^{co}$, twice the formation energy for one A-Ag-P node (see Supplementary Discussion for details). As a result, the reaction energies are calculated to be $E_r^{homo}$ = -2.24 eV and $E_r^{co}$ = -2.29 eV per A* moiety involved in the reaction, showing that the energy gain upon copolymerization is 0.05 eV larger than that upon homopolymerizations. Such a small energy difference for each A* involved accumulates as the surface-confined

polymerization proceeds, and hence would result in a remarkable preference for the copolymerization product when the reaction equilibrium is established due to the exponential dependence of the population of different products on their energy difference, according to the Boltzmann distribution. As a result, the formation of alternating copolymer is thermodynamically preferred in the reaction system at $r > 1$. This explains the selective formation of the alternating [A-Ag-P-Ag]$_n$ product at $r = 2$, i.e., the P* involved in the reaction is twice as A*, when a half of the P* residues are consumed by the activation of A-H and the other half copolymerize with the activated A*. At $r > 2$, the large amount of P-Br would facilitate the formation of the copolymeric product due to a more feasible approach of the activated A* residues to the P* ones. The excessive P* species that are not engaged in the copolymerization then undergo homopolymerization to form the [P-Ag]$_n$ product. Based on these analyses, the reaction pathways between A-H and P-Br on Ag(111) at varied $r$ intervals are schematically shown in Fig. 6b. The accordingly achieved overall reactions in the surface-confined system are summarized in Table 1, and those of the [A-Ag]$_n$ + P-Br and [P-Ag]$_n$ + A-H reaction systems are provided in Supplementary Table 1.

A quantitative description of the $r$-dependence of the polymerization selectivity can be given by definition of the yields of the [A-Ag]$_n$ ($Y_{[A-Ag]_n}$), [A-Ag-P-Ag]$_n$ ($Y_{[A-Ag-P-Ag]_n}$) and [P-Ag]$_n$ ($Y_{[P-Ag]_n}$) products as the proportions of the intermolecular A-Ag-A, A-Ag-P and P-Ag-P connections to the total amount of the organometallic nodes in the system, correspondingly: for $0 < r \leq 1$,

$$\begin{cases} Y_{[A-Ag]_n} & = 100\% \\ Y_{[A-Ag-P-Ag]_n} & = 0 \\ Y_{[P-Ag]_n} & = 0 \end{cases}, \qquad (1)$$

for $1 < r \leq 2$,

$$\begin{cases} Y_{[A-Ag]_n} & = \frac{2-r}{r} \times 100\% \\ Y_{[A-Ag-P-Ag]_n} & = \frac{2(r-1)}{r} \times 100\% , \\ Y_{[P-Ag]_n} & = 0 \end{cases} \qquad (2)$$

and for $r > 2$,

$$\begin{cases} Y_{[A-Ag]_n} & = 0 \\ Y_{[A-Ag-P-Ag]_n} & = \frac{2}{r} \times 100\% . \\ Y_{[P-Ag]_n} & = \frac{r-2}{r} \times 100\% \end{cases} \qquad (3)$$

The as-achieved yields of the organometallic products as a function of $r$ are depicted by the solid curves in Fig. 6c. They are compared with the experimental data estimated from the samples with $r$ ranging from 0.6 to 6.5, as plotted by the scatters in Fig. 6c (see Supplementary Discussion for details). The good consistency between the computational and experimental results supports our proposed reaction mechanism. These yield-$r$ relationships are valid for both the polymerization between two monomers and the homo- to co-copolymerization transformations.

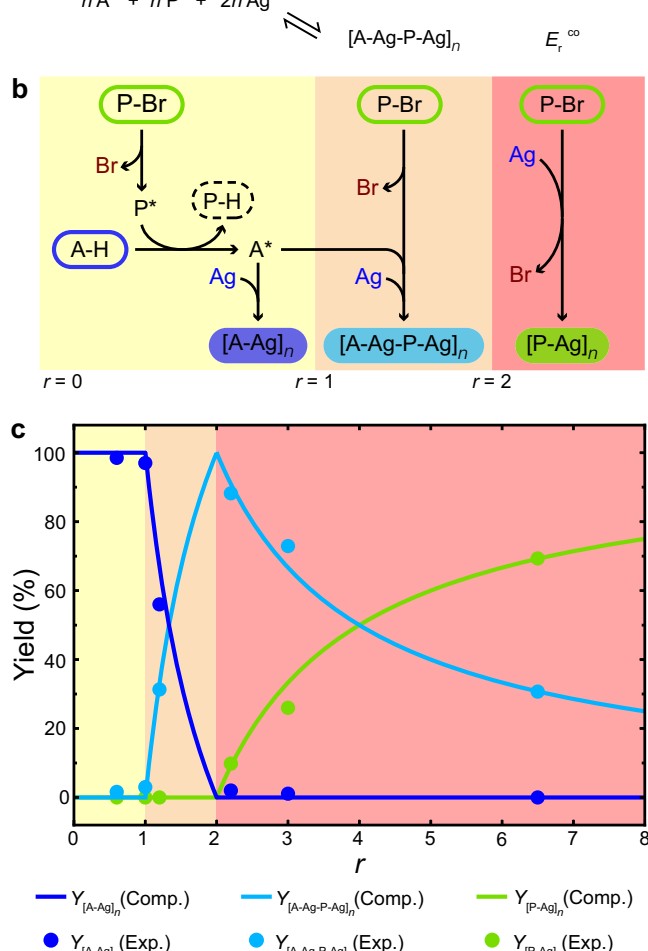

**Fig. 6 | Reaction pathways and $r$-dependent polymerization selectivity.** **a** Competing homo- and co-polymerizations. **b** Schematic illustration of the surface reaction pathways between A-H and P-Br. **c** Computational and experimental yields of the [A-Ag]$_n$, [A-Ag-P-Ag]$_n$ and [P-Ag]$_n$ products as a function of $r$. The regions of $0 < r \leq 1$, $1 < r \leq 2$ and $r > 2$ in the graph are shaded in yellow, orange and red, correspondingly.

**Table 1 | Overall Reactions between A-H and P-Br at Varied $r$ Intervals**

| $r$ | Overall Reaction |
|---|---|
| $0 < r \leq 1$ | $n$ A-H + $n$ P-Br + $n$ Ag → [A-Ag]$_n$ + $n$ Br + $n$ P-H |
| $1 < r \leq 2$ | $n$ A-H + $rn$ P-Br + $rn$ Ag → $(r-1)$ [A-Ag-P-Ag]$_n$ + $(2-r)$ [A-Ag]$_n$ + $rn$ Br + $n$ P-H |
| $r > 2$ | $n$ A-H + $rn$ P-Br + $rn$ Ag → [A-Ag-P-Ag]$_n$ + $(r-2)$ [P-Ag]$_n$ + $rn$ Br + $n$ P-H |

$r$ and $n$ refer to the P-Br: A-H ratio and the number of repeating units in the polymers, respectively.

## Discussion

The crucial role played by the Ag adatoms in the sequence-controlled copolymerization is twofold. On the one hand, the reversibility of the reactions between the Ag adatoms and the A*/P* residues enables the thermodynamic control of the final products. As a thermodynamic strategy, the change in the stoichiometric ratio of the monomers can efficiently tune the polymerization selectivity. On the other hand, the difference in the formation energies of the Ag-involved organometallic species, i.e., A-Ag-A, P-Ag-P and A-Ag-P, determines the polymerization equilibrium and hence the final organometallic product. This leads to the preferential generation of the copolymeric product once both reactive A* and P* are available on the surface ($r > 1$) because the formation of the A-Ag-P node is more energetically favorable.

The stoichiometric effect on the surface-confined homo-/copolymerization selectivity was also elucidated in another Ag(111)-supported system composed of different aromatic bromide and terminal alkyne monomers, i.e., 4,4″-dibromo-1,1′:3′,1″-terphenyl and 1,2-bis(4-ethynylphenyl)ethyne (Supplementary Fig. 6a–d). As a result, their on-surface organometallic polymerization presented an $r$-dependent selectivity similar to that between P-Br and A-H (see Supplementary Discussion for details). The system underwent selective homo-polymerization of the alkyne monomers at $r \leq 1$ (Supplementary Fig. 6e, f), and the homopolymerization of the bromides prevailed when the bromides were in far excess to the alkynes (Supplementary Fig. 6i, j). As a comparison, the two monomers preferentially copolymerized into the sequence-controlled organometallic chains at middle stoichiometric ratios (Supplementary Fig. 6g, h). The similar reaction scenario in both Ag(111)-supported polymerization systems composed of different terminal alkynes and aromatic bromides suggests that the stoichiometric control be a promising strategy to tune the homo-/copolymerization selectivity in these systems.

To conclude, the selectivity of the organometallic polymerizations of coexistent A-H and P-Br on Ag(111) was effectively tweaked by stoichiometric control of these two monomers. The stoichiometric control was achieved by either stoichiometric dosage of the monomers at the initial stage of the reaction, or post-addition of one monomer to the surface pre-covered by homopolymers composed of the other one. As a result, highly selective alternating copolymerization was achieved at $r = 2$. The sequence-controlled organometallic polymerization may not only improve the mechanical and electronic properties of the organic polymers, but also achieve promising functionalities such as conductivity, non-linear optics, catalysis and so on and so forth[47]. This study paves a way to molecularly precise sequence control of copolymerization via surface chemistry, which should register great applications for precise fabrication of complicated polymeric nanostructures by on-surface synthesis.

## Methods

### STM experiments

All STM experiments were performed with a Unisoku UHV-STM with a background pressure below $3 \times 10^{-10}$ Torr. The atomically flat Ag(111) surface was prepared by repeated cycles of Ar$^+$ sputtering and annealing at about 780 K. Both A-H and P-Br molecules were evaporated from the tantalum boats at about 420 K and 450 K, respectively, onto the substrate held below RT. The surface-confined organometallic reactions can be initiated by thermal treatments of the samples at mild temperatures ranged from RT to 340 K. Specifically, the annealing at RT was usually conducted for >10 h to ensure the establishment of the reaction equilibrium. All STM images were acquired at 4.9 K in constant current mode and processed by the WSxM software[48].

### Calculations

All first-principles calculations were performed based on DFT as implemented in the Vienna ab initio simulation package (VASP)[49,50].

The projector augmented-wave (PAW) method was used to describe the electron-ion interaction[51]. The Perde-Burke-Ernzerhof (PBE) form of generalized gradient approximation (GGA) was used to describe the electron exchange and correlation energy[52]. The DFT long-range dispersion correction (DFT-D3) method was adopted for vdW correction to the PBE functional[53,54]. For the model construction, the Ag(111) surface was described by a slab of 168 atoms comprising 4 atomic layers. The dimension of the molecule-slab super-cell was 20.1 Å × 15.0 Å × 22.0 Å. The molecules were fully relaxed on the Ag(111) substrate with two bottom layers fixed until the residual forces were <0.02 eV/Å. The kinetic energy cut-off was 500 eV, and the convergence criteria for total energy was $10^{-5}$ eV. The sampling of the Brillouin zone was performed using a Monkhorst-Pack scheme by the set of $1 \times 1 \times 1$ k-point. Calculations for the potential pathway of the dehydrogenation of phenylacetylene were conducted with Climbing Image Nudged-Elastic Band (CI-NEB) method as implemented in VASP through the VTST-Tools[55,56]. The convergence criterium of ionic steps for the NEB calculations was 0.03 eV/Å.

## Data availability

The data that support the findings of this study are available within the paper, supplementary information and source data file. All data are available from the corresponding author upon request. Source data are provided with this paper.

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

## Acknowledgements

This work is jointly supported by NSFC [92356309 (Y.W., J.Liu., and Q.C.), 21821004 (K.W.), 21932001 (K.W.), 22101008 (Q.C.), 22372003 (J.Liu.) and 22225202 (Y.W.)]. DFT calculations are carried out on TianHe-1A at National Supercomputer Center in Tianjin and supported by High-Performance Computing Platform of Peking University and the Beijing Super Cloud Computing Center (BSCC) (URL: htP-H://www.blsc.cn/).

## Author contributions

K.W. initiated the project, analyzed the data and wrote the manuscript; L.Xin., Y.B. and Y.L. conducted the STM experiments; J.Liu and Q.C. analyzed the data and wrote the manuscript; J.Li, and C.L. carried out DFT calculations; Y.W. analyzed the data and participated in DFT calculations; L.Xiao and D.Z. synthesized the monomers.

## Competing interests

The authors declare no competing interests.
