## [Peer Review File · Nature Communications]

Surface-Confined Alternating Copolymerization with
Molecular Precision by Stoichiometric ControlReviewers' Comments:

Reviewer #1:

Remarks to the Author:

The present study employs LT- STM to investigate the process of polymerization involving two types of molecules possessing bromine and alkynyl moieties on the Ag(111) surface. A variety of stoichiometric ratios of these two molecules are deposited onto the Ag(111) surface and subsequently subjected to annealing procedures. Remarkably, the research found that the polymerization is amenable to regulation through manipulation of the stoichiometry ratio. Furthermore, the study establishes the feasibility of achieving alternating copolymerization, particularly evidenced by a specific ratio of 2. The synthesis of polymers with well-defined block periodicity hold significance within chemistry. The attainment of such synthesis with high yield constitutes a substantial challenge in its own right. Nevertheless, a few inquiries arise concerning the precise mechanisms underpinning the reaction processes and corresponding explanations. Thus, before considering for publication, some concerns listed in the following should be addressed.

1. Based on the topography images, one would expect that the stoichiometry ratio can only be roughly defined on a very large scale instead of microscopically. Intuitively, one would expect that the phenomena be explained by the formation energy of the different types of polymers. It will be nice to investigate the formation energies in a bit more detail and comment on how these two physics pictures consistent with each other.
2. It is known that STM is a technique mainly used to probe system at the static(equilibrium) states. It is hard to comment on the dynamic chemical reaction processes during annealing. Nevertheless, I am wondering if the authors can provide some microscopic reaction path pathway for the alternating copolymerization in both the co-deposition and post-deposition cases?
3. I suppose one needs to suppose that the reaction sufficiently efficient and complete during the RT annealing process. What is the annealing period and how it is determined? Similarly, why RT is used for the annealing temperature?
4. In the introduction, the authors mentioned the case of DNA, there are more complex structures can be formed. Have the authors observed more complex structures? If yes, why it is reasonable to exclude them in the discussion?
5. The effect of the Ag adatoms in the copolymerization should be discussed in a bit more detail. Does its activity affect the stoichiometry ratio?
6. How general is the results on similar systems? The ratio is an integer number 2, is it accidental or valid for similar systems, such as other metal surfaces? And what is its limitation, is this ratio depending on other parameters such as annealing temperature?
7. What are the coverage ratios per unit area for the different types of polymerization cases? What are the effects of the inter-chain coupling in the process?
8. Can the authors provide some large-scale topographic images to get better views of the surface homogeneity?

Reviewer #2:

Remarks to the Author:

The manuscript by L. Xing and co-workers reports on a very interesting case of on-surface polymerization and stoichiometric control, as studied by scanning tunneling microscopy and DFT calculations. The monomers 4,4''-dibromo-p-terphenyl (P-Br) and 2,5-diethynyl-1,4-bis(phenylethynyl)benzene (A-H) with Ag adatoms on Ag(111) surface form the phenyl-silver-phenyl (P-Ag-P) and alkynyl-silver-alkynyl (A-Ag-A) intermolecular organometallic connections via the dehalogenated and dehydrogenated metalation reactions. The manuscript is well written, and the data presented are of high quality. Overall, the manuscript is interesting but would require minor revisions and I have only three concerns in relation to this study.

- 1) The main point of this work is stoichiometric control of the polymerization by stoichiometric dosage

with high selectivity. Does the stoichiometric control only work for the specific precursor? Is it applicable for different on-surface systems with different monomers? If possible experimental evidence on the universality by using slightly modified monomer should be provided to get further insight.

2) In general, on-surface synthesis sensitively depends on the temperature of substrate. why not author studied on surface synthesis by varying temperature? Also, I have concern about simulation that What is the temperature in simulation, room temperature?

Reviewer #3:

Remarks to the Author:

Xing and coauthors reported the selective alternating copolymerization between an aromatic halide and a terminal alkyne on Ag(111) by combined STM and DFT studies. The authors explored the surface organometallic polymerization selectivity by the stoichiometric control of two components and claimed that selectivity preference of hetero-intermolecular organometallic connection is achieved at stoichiometric ratio of 2. The thermodynamic control of the homo-/co-polymerization is realized by tuning of the stoichiometric ratio both experimentally and theoretically. Totally this is an interesting study which certainly contributes to the emerging "on-surface chemistry" field. The idea is original and the manuscript is well written. However, several major concerns as listed below prevent me from recommending its publication in Nature Communications at this stage.

1) The background topic is the sequence-controlled copolymerization, which is of importance in synthetic chemistry. Within the field of on-surface chemistry, this work deals with the selective alternating organometallic copolymerization via intermolecular metalation of two components with Ag atoms on Ag(111) surfaces. The lack of discussion on the potential application/transferring from controlled organometallic reactions to real covalent polymerization significantly reduces its novelty and applicability.

2) To clarify the specific P-Br-related species that takes effect on promoting the C-H bond activation in A-H, the authors performed a series of control experiments. With regard to the reaction of A-H with $[P-Ag]_n$, in Fig. 4b and 4d, STM data verified the efficient activation of the alkynyl C-H bond in A-H at RT. In the later discussion, authors attributed it to P^* which deviates with the dissociation of P-Ag species. However, based on the reaction temperature (400 K) and calculated energy barrier (0.88 eV) of C-Ag dissociation in the cited paper (Angew. Chem. Int. Ed. 56, 12852-12856 (2017)), P^* is not available in the reaction system of A-H with $[P-Ag]_n$ at RT. Please comment on this concern.

3) In the discussion part of competition between homo-polymerization and co-polymerization, the reaction mechanism is explained in detail by comparing reaction energies, rationalizing the selective formation of the alternating $[A-Ag-P-Ag]_n$ product at $r = 2$. But when $r > 2$, it seems that for A-H there will always be 100% formation yield of $[A-Ag-P-Ag]_n$, although the overall yield will decrease as excess P-Br increases. Is that obvious, or does the increase of P-Br have other side effects when $r > 2$.

4) The universality of the stoichiometric control as a synthetic strategy is not confirmed. Does the findings only work for the specific (halide and alkyne) system? Prospect on the universality of the strategy can be addressed.

5) In addition, there are two minor issues the authors should address.

a) Since the stoichiometric ratio between two monomers is the main parameter to steer the surface organometallic polymerization selectivity, the authors should describe the precise way to determine the ratio. Or it is roughly counted from STM images?

b) What is the temperature of substrates when initially co-depositing molecules on surfaces, for example, in Fig. 2. And what is the RT annealing time to initiate the surface-confined organometallic (co)polymerization.

c) How do authors calculate the experimental yields in Fig. 6c.

Dear Reviewers,

Thank you for your insightful advice and comments that help us improve the quality of the manuscript entitled “Surface-Confined Alternating Copolymerization with Molecular Precision by Stoichiometric Control”, submitted for publication as an article in *Nature Communications*.

We have carefully considered all your comments/suggestions and conducted additional new experiments and revised our manuscript accordingly. As the concern about the universality of the stoichiometric strategy has been raised by all three Reviewers, in the following, we'll firstly present the relevant new experimental results and the corresponding revisions. Then, we'll address one by one your comments and questions. All the responses are typed in blue. All changes in the manuscript are marked in red (also highlighted in red in the traced version of the revised manuscript, supplied as Review-Only file) so that you can feasibly identify the whereabouts of our revisions.

Thank you for your reconsideration.

Sincerely,

Kai Wu (On behalf of all co-authors)

Table of Contents

1. Additional Experimental Results	R3
2. Point-to-Point Replies	R5
Reply to Reviewer #1	R5
Reply to Reviewer #2	R13
Reply to Reviewer #3	R15
3. Other Non-Scientific Revisions	R20

1. Additional Experimental Results

We have studied the reaction between different aromatic bromide and terminal alkyne monomers on Ag(111). The results reveal a similar reaction scenario in the new system as that demonstrated between A-H and P-Br in the main text. The details are added to the Supplementary Materials on Pages S9-S11 as below:

“4. Reaction of 1,2-Bis(4-ethynylphenyl)ethyne and 4,4''-Dibromo-1,1':3',1''-terphenyl on Ag(111)

Co-evaporation of 4,4''-dibromo-1,1':3',1''-terphenyl (upper in Supplementary Fig. 6a, denoted as P'-Br) and 1,2-bis(4-ethynylphenyl)ethyne (lower in Supplementary Fig. 6a, denoted as A'-H) onto the Ag(111) substrate held below RT led to their self-assemblies into separate domains, as shown in Supplementary Fig. 6b-d.

Supplementary Fig. 6 Surface-confined reaction between P'-Br and A'-H. (a) Chemical structures of P'-Br and A'-H. (b) STM image of the co-existent self-assembly domains of P'-Br and A'-H ($V = 1$ V and $I = 100$ pA). Scale bar: 10 nm. High-resolution STM images of the self-assemblies of (c) P'-Br ($V = 300$ mV and $I = 200$ pA) and (d) A'-H ($V = 300$ mV and $I = 200$ pA). Scale bars: 2 nm. Large-area STM images of the reaction products achieved by P'-Br and A'-H on Ag(111) at (e) $r \sim 0.7$ ($V = 1$ V and $I = 100$ pA), (g) $r \sim 3$ ($V = 300$ mV and $I = 200$ pA), and (i) $r \sim 8$ ($V = 300$ mV and $I = 200$ pA). Scale bars: 10 nm. Inset of (e): High-resolution STM image of the co-assembly formed by P'-H

and Br adatoms ($V = 300$ mV and $I = 200$ pA). Scale bar: 1 nm. High-resolution STM images of (f) the $[A'-Ag]_n$ chains ($V = 100$ mV and $I = 200$ pA), (h) the $[A'-Ag-P'-Ag-P'-Ag]_n$ chains ($V = 600$ mV and $I = 200$ pA), and (j) the $[P'-Ag]_n$ chains ($V = 300$ mV and $I = 200$ pA). The extending orientations of the chains are highlighted by the colored dashed lines. The chemical structures of the polymeric products are illustrated in bottom panels of (f), (h) and (j). Scale bars: 1 nm for (f) and (j), and 2 nm for (h).

An r -dependent ($r = P'-Br : A'-H$) homo-/co-polymerization selectivity was noticed in the $A'-H + P'-Br$ system after annealing of the samples pre-covered by the two monomers at varied stoichiometric ratios at RT - 340 K. On the samples at $r \leq 1$ (e.g., $r \sim 0.7$, Supplementary Fig. 6e), the homopolymeric $[A'-Ag]_n$ chains (shaded in dark blue) appeared as the main product. The assembled $[A'-Ag]_n$ chains coexisted with less ordered domains (marked by the dashed black frame in Supplementary Fig. 6e) formed by 1,1':3',1''-terphenyl (denoted as P'-H, highlighted by the dashed black frame in the inset) and the detached Br adatoms (marked by the dashed white circles). P'-H is presumably the byproduct due to the passivation of the debrominated P'-Br by the H atoms produced from the A-H dehydrogenation, according to our mechanistic analyses in the main text. A high-resolution STM image of the $[A'-Ag]_n$ chains is given in Supplementary Fig. 6f, showing the alternate arrangement of the rod-like molecular moieties and the circular protrusions assigned as the Ag adatoms.

Once P'-Br is in far excess to A'-H (e.g., $r \sim 8$, Supplementary Fig. 6i), the surface of the annealed sample was dominantly covered by the zigzag chains (shaded in green) that were identified as the $[P'-Ag]_n$ homopolymers according to previous reports^{2,3}. A closeup of the zigzag chains (Supplementary Fig. 6j) resolves the alternate arrangement of the 120° V-shaped molecular moieties and the Ag adatoms, i.e., the round and bright protrusions. Detached Br adatoms appear as the dim dots located in between the $[P'-Ag]_n$ chains, as marked by the dashed white circles in Supplementary Fig. 6j. In addition to the main product $[P'-Ag]_n$, the copolymeric structures (discussed below) were also obtained, as shaded in light blue in Supplementary Fig. 6i.

At middle stoichiometric ratios, copolymers with an ordered sequence (shaded in light blue) became the main product on the surface, in coexistence with the less ordered structures composed of randomly aligned organometallic chains and the P'-H molecules. The high-resolution STM image of the copolymers (Supplementary Fig. 6h) reveals a $[A'-Ag-P'-Ag-P'-Ag]_n$ sequence, indicating that the product is an “ABB” type copolymer. The yield of the $[A'-Ag-P'-Ag-P'-Ag]_n$ copolymer maximizes at $r \sim 3$ (Supplementary Fig. 6g). The dim dots located in between the copolymeric chains, as marked by the dashed white circles in Supplementary Fig. 6h, are the detached Br adatoms.

The on-surface reaction between P'-Br and A'-H shows an r -dependent homo-/co-polymerization selectivity similar to that between P-Br and A-H. When the amount of the bromide is fewer than ($r \leq 1$) and far in excess to that of the alkyne, the homopolymers $[A-Ag]_n$ (or $[A'-Ag]_n$) and $[P-Ag]_n$ (or $[P'-Ag]_n$) become the main products, respectively. At middle ratios, the copolymers with ordered sequences prevail on the surface. These findings suggest the similarities in two aspects of the reaction mechanisms for both systems. One is the activation of the alkyne monomers consumes the bromides of equivalent amount following the reaction,

which explains the selective homopolymerization of A-H (or A'-H) to give $[\text{A-Ag}]_n$ (or $[\text{A'-Ag}]_n$) at $r \leq 1$ and the appearance of the P-H (or P'-H) byproduct. The other is the preferred copolymerization rather than the homopolymerization once both A^* (or A'^*) and P^* (or P'^*) are available in the reaction systems.

In terms of the “ABB” sequence of the copolymeric product formed by A'-H and P'-Br, our speculation is twofold. On the one hand, the chain periodicity of the $[\text{A'-Ag-P'-Ag-P'-Ag}]_n$ copolymer is in commensuration with the substrate lattice along the orientations the chains extend. Such a chain-substrate commensurability may selectively stabilize the ABB-type copolymer. On the other hand, the formation of the well-ordered and close-packed assemblies of the $[\text{A'-Ag-P'-Ag-P'-Ag}]_n$ chains (Supplementary Fig. 6g and h) may also contribute to their stabilization.”

We also add a paragraph to the main text (Paragraph 2, Page 20) to summarize and discuss the results as below:

“The stoichiometric effect on the surface-confined homo-/co-polymerization selectivity was also elucidated in another Ag(111)-supported system composed of different aromatic bromide and terminal alkyne monomers, i.e., 4,4"-dibromo-1,1':3,1"-terphenyl and 1,2-bis(4-ethynylphenyl)ethyne (Supplementary Fig. 6a-d). As a result, their on-surface organometallic polymerization presented an r -dependent selectivity similar to that between P-Br and A-H (see Supplementary Discussion for details). The system underwent selective homopolymerization of the alkyne monomers at $r \leq 1$ (Supplementary Fig. 6e and f), and the homopolymerization of the bromides prevailed when the bromides were in far excess to the alkynes (Supplementary Fig. 6i and j). As a comparison, the two monomers preferentially copolymerized into the sequence-controlled organometallic chains at middle stoichiometric ratios (Supplementary Fig. 6g and h). The similar reaction scenario in both Ag(111)-supported polymerization systems composed of different terminal alkynes and aromatic bromides suggests that the stoichiometric control be a promising strategy to tune the homo-/co-polymerization selectivity in these systems.”

2. Point-to-Point Replies

Reviewer #1

The present study employs LT- STM to investigate the process of polymerization involving two types of molecules possessing bromine and alkynyl moieties on the Ag(111) surface. A variety of stoichiometric ratios of these two molecules are deposited onto the Ag(111) surface and subsequently subjected to annealing procedures. Remarkably, the research found that the polymerization is amenable to regulation through manipulation of the stoichiometry ratio. Furthermore, the study establishes the feasibility of achieving alternating copolymerization, particularly evidenced by a specific ratio of 2. The synthesis of polymers with well-defined block periodicity hold significance within chemistry. The attainment of such synthesis with high yield constitutes a substantial challenge in its own right. Nevertheless, a few inquiries arise concerning the precise mechanisms underpinning the reaction processes and corresponding explanations. Thus, before considering for publication, some concerns listed in the following should be addressed.

1. Based on the topography images, one would expect that the stoichiometry ratio can only be roughly defined on a very large scale instead of microscopically. Intuitively, one would expect that the phenomena be explained by the formation energy of the different types of polymers. It will be nice to investigate the formation energies in a bit more detail and comment on how these two physics pictures consistent with each other.

Author reply:

We greatly appreciate the Reviewer for the overall positive and suggestive comments of our manuscript. We'd like to reply to the comment in two parts.

(1) Determination of stoichiometric ratios

We have added a paragraph in the Supplementary Materials (Paragraph 1, Page S8) to describe how we determine the stoichiometric ratios:

“1. Determination of Stoichiometric Ratios

The stoichiometric ratio between the P-Br and A-H monomers was experimentally controlled by their evaporation times at specific evaporation fluxes, and then confirmed by exhausting counting of both monomer numbers on the sampled surface. The STM images used for the statistical analyses are acquired at random spots of the substrate surface. The ratio value, r , is calculated by the weighted average of the P-Br molecular density in the assembled and disordered structures on the surface divided by that of A-H. The calculated molecular density of an ordered molecular assembly is the number of the molecules within a unit cell divided by the unit cell area. The molecular density of a less ordered zone is estimated by the total number of the molecules divided by the zone area that is experimentally monitored.”

The phrases referring to the statement has been added to the main text (Paragraph 2, Page 6):

“The attempt aiming at selective alternating copolymerization began with the reaction between the P-Br and A-H monomers at a ratio of $r \sim 1$ (details for r determination are provided in the Supplementary Discussion) on Ag(111) ...”

(2) Discussion on Formation Energies

In the manuscript, we explain the selective copolymerization,

as a result of its lower reaction energy per A^* involved ($E_r^{CO} = -2.29$ eV) than that of homopolymerization ($E_r^{homo} = -2.24$ eV),

Detailed discussions can be found in the main text (Pages 16-18) and Supplementary Materials (Page S8). The comparison between the reaction energies is essentially that between the formation energy per unit for $[A-Ag-P-Ag]_n$ and the summation of those for $[A-Ag]_n$ and $[P-Ag]_n$. These polymer formation energies are estimated by corresponding formation energies for two A-Ag-P nodes ($2E_f^{A-Ag-P}$), one A-Ag-A node (E_f^{A-Ag-A}) and one P-Ag-P node (E_f^{P-Ag-P}). Therefore, the relationship between the reaction and formation energies can be expressed as $E_r^{CO} = 2E_f^{A-Ag-P}$ and $E_r^{homo} = E_f^{P-Ag-P} + E_f^{A-Ag-A}$. Phrases and sentences have now been added to the main text and Supplementary Materials to clarify the relationship between the reaction and formation energies.

Paragraph 1, Page 17 in the main text: “... the reaction energies of homopolymerization (E_r^{homo})

and copolymerization (E_r^{co}) between A* and P* (Fig. 6a) are compared. E_r^{homo} is estimated by summation of the formation energies for one P-Ag-P node and one A-Ag-A node, and E_r^{co} , twice the formation energy for one A-Ag-P node (see Supplementary Discussion for details). As a result, the reaction energies are calculated to be $E_r^{homo} = -2.24$ eV and $E_r^{co} = -2.29$ eV per A* moiety involved in the reaction, ...”

Page S8 in the Supplementary Materials: “ E_r^{homo} and E_r^{co} refer to the reaction energies per A* involved of the polymerization Reactions (1) and (2), respectively, as listed below:

The reaction energy for Reaction (1) can be estimated by summation of the formation energies for one P-Ag-P (E_f^{P-Ag-P}) and one A-Ag-A (E_f^{A-Ag-A}) intermolecular organometallic connections,

$$E_r^{homo} = E_f^{P-Ag-P} + E_f^{A-Ag-A} = E_{P-Ag-P/Ag(111)} + E_{A-Ag-A/Ag(111)} - 2E_{P^*/Ag(111)} - 2E_{A^*/Ag(111)} - 2E_{Ag/Ag(111)} + 4E_{Ag(111)},$$

while that for Reaction (2), twice the formation energy for one A-Ag-P node (E_f^{A-Ag-P}),

$$E_r^{co} = 2E_f^{A-Ag-P} = 2(E_{A-Ag-P/Ag(111)} - E_{P^*/Ag(111)} - E_{A^*/Ag(111)} - E_{Ag/Ag(111)} + 2E_{Ag(111)}).$$

The formation energies for the organometallic nodes are calculated as:

$$E_f^{P-Ag-P} = E_{P-Ag-P/Ag(111)} - 2E_{P^*/Ag(111)} - E_{Ag/Ag(111)} + 2E_{Ag(111)},$$

$$E_f^{A-Ag-A} = E_{A-Ag-A/Ag(111)} - 2E_{A^*/Ag(111)} - E_{Ag/Ag(111)} + 2E_{Ag(111)},$$

$$\text{and } E_f^{A-Ag-P} = E_{A-Ag-P/Ag(111)} - E_{A^*/Ag(111)} - E_{P^*/Ag(111)} - E_{Ag/Ag(111)} + 2E_{Ag(111)},$$

where $E_{P-Ag-P/Ag(111)}$, $E_{A-Ag-A/Ag(111)}$ and $E_{A-Ag-P/Ag(111)}$ correspondingly refer to the energies for the P-Ag-P, A-Ag-A and A-Ag-P dimers adsorbed on Ag(111). $E_{Ag/Ag(111)}$, $E_{P^*/Ag(111)}$, and $E_{A^*/Ag(111)}$ correspond to the energy for one Ag adatom, one P* residue, and one A* residue on Ag(111). $E_{Ag(111)}$ is the energy of the Ag(111) substrate. The as-achieved formation energies for the organometallic species are now added to the Supplementary Materials as Supplementary Table 2 on Page S7 as below:

Supplementary Table 2. Formation Energies for the Organometallic Nodes.

Structure	Formation Energy (eV)
A-Ag-A	-0.940
P-Ag-P	-1.299
A-Ag-P	-1.144

One sentence referring to the table is added to the Supplementary Materials (Paragraph 1, Page S9) as below:

“The accordingly achieved formation energies for the organometallic structures are listed in Supplementary Table 2.”

The energetic analyses indicate the alternating copolymerization to be more energetically favorable than separate homopolymerizations of A-H and P-Br. Accordingly, the r -dependent polymerization selectivity can be anticipated as follow, which is consistent with the experimental findings (detailed discussion can be found in the main text on Pages 16-19). At $r \leq 1$, the P* residues originated from the P-Br debromination are completely consumed by the activation of A-H and

irreversibly converted into the P-H byproduct. Therefore, only the A* residues generated by the dehydrogenated activation of A-H are available for the organometallic polymerization, yielding $[A-Ag]_n$ as the only polymeric product. At $r > 1$, both A* and P* in addition to those consumed by the A-H activation are available. Therefore, $[A-Ag-P-Ag]_n$ is energetically favorable. Besides the copolymerization, excessive A* (at $1 < r < 2$) and P* (at $r > 2$) would conduct homopolymerization to form $[A-Ag]_n$ and $[P-Ag]_n$, respectively. Hence, both $[A-Ag]_n$ and $[A-Ag-P-Ag]_n$ are produced at $1 < r < 2$, while $[A-Ag-P-Ag]_n$ and $[P-Ag]_n$ coexist at $r > 2$. Only if $r = 2$, the reactive A* and P* residues become equalized in amount (a half of the P* residues being passivated upon their reaction with A-H), and the alternating copolymerization takes place to produce the dominant product, $[A-Ag-P-Ag]_n$.

2. *It is known that STM is a technique mainly used to probe system at the static (equilibrium) states. It is hard to comment on the dynamic chemical reaction processes during annealing. Nevertheless, I am wondering if the authors can provide some microscopic reaction path pathway for the alternating copolymerization in both the co-deposition and post-deposition cases?*

Author reply:

Yes, the Reviewer is right that it's normally hard to monitor the chemical dynamics during annealing processes with low-temperature STM. However, it has become a routine approach that the ongoing reaction can be quickly frozen prior to the reaction equilibrium, and the molecular species formed on surface at the early reaction stages can be imaged and diagnosed by STM.

Fig. R1 (a) STM image of the A-H + P-Br sample at $r \sim 2$ after being annealed at RT for 4 hours ($V = 100$ mV and $I = 100$ pA). The $[A-Ag]_n$ chains are shaded in dark blue. STM images of (b) the $[P-Ag]_n + A-H$ ($V = 500$ mV and $I = 100$ pA) and (c) $[A-Ag]_n + P-Br$ ($V = 10$ mV and $I = 100$ pA) samples at their early reaction stages. The A-Ag-P nodes at the endings of the polymeric chains are marked by the light blue arrows. Scale bars: (a) and (c) 8 nm, (b) 5 nm.

Fig. R1a shows a representative STM image of the sample surface pre-covered by P-Br and A-H monomers at $r \sim 2$ after its annealing at room temperature (RT) for 4 hours. Note that the complete conversion of the monomers into the alternating copolymers, as shown in Fig. 3e in the main text, can be clearly observed on the sample after its annealing at RT for 15 hours in total. Therefore, Fig. R1a pictorially illustrates the early reaction stage. One can actually identify the $[A-Ag]_n$ chains in Fig. R1a (shaded in dark blue) in coexistence with the unreacted P-Br monomers randomly adsorbed on

the surface. This experimental observation suggests that the $[A-Ag]_n$ species be the intermediate of the final reaction product $[A-Ag-P-Ag]_n$. Therefore, the alternating copolymerization in the co-evaporation case mainly proceeds in two steps. Firstly, the A-H monomers homopolymerize into $[A-Ag]_n$ after being activated via their reaction with the debrominated P* residues (please refer to the detailed discussion on Pages 12-16 in the main text). Secondly, the formed $[A-Ag]_n$ chains react with the P-Br monomers to yield the final product, $[A-Ag-P-Ag]_n$.

Likewise, the STM images of the samples at the early reaction stages in the post-evaporation cases, including $[P-Ag]_n + A-H$ and $[A-Ag]_n + P-Br$, can also be acquired, as depicted in Fig. R1b and c, respectively. One can feasibly identify the A-Ag-P nodes (marked by the light blue arrows in Fig. R1b and c) at the endings of the $[A-Ag]_n$ and $[P-Ag]_n$ chains, indicating that the homo- to copolymerization transformation starts from the endings of the homopolymeric chains.

3. *I suppose one needs to suppose that the reaction sufficiently efficient and complete during the RT annealing process. What is the annealing period and how it is determined? Similarly, why RT is used for the annealing temperature?*

Author reply:

To ensure the establishment of the reaction equilibrium, the samples are typically annealed at RT for more than 10 hours. For example, the A-H + P-Br sample at $r \sim 2$ that gave rise to the alternating copolymers was annealed at RT for 15 hours in total. This information is now added to the “Methods” section on Page 21 in the main text as below:

“Specifically, the annealing at RT was usually conducted for more than 10 hours to ensure the establishment of the reaction equilibrium.”

The completeness of the reaction is confirmed by the absence of either unreacted monomers on the surface or substantial change of the molecular structures after further thermal treatment of the sample at RT or slightly higher (e.g., 340 K).

We have varied the annealing temperatures to show that the stoichiometric ratio controlled organometallic polymerization can take place at mild temperatures ranged from RT to 340 K. The reactions proceed at a moderate rate at RT, which favors the precise control of the reaction process. Therefore, most experimental results are acquired by annealing of the samples at RT and the as-achieved data are presented in the manuscript. We have now added one sentence to the “Methods” section on Page 21 to clarify the reaction temperature as below:

“The surface-confined organometallic reactions can be initiated by thermal treatments of the samples at mild temperatures ranged from RT to 340 K.”

Further thermal annealing at an even higher temperature would cause unavoidable less-ordered intermolecular covalent coupling of the surface species. Such irreversible reactions are kinetically controlled and hence the stoichiometric strategy (i.e., a thermodynamic effect) for polymerization selectivity regulation would become inapplicable. Therefore, we typically conduct annealing of the samples at around RT or slightly higher.

4. *In the introduction, the authors mentioned the case of DNA, there are more complex structures can be formed. Have the authors observed more complex structures? If yes, why it is reasonable*

to exclude them in the discussion?

Author reply:

We thank the Reviewer for the comment. We have observed less ordered organometallic chains composed of randomly arranged homo- and co-polymeric segments mainly on the samples at $1 < r < 2$ and $r > 2$. The examples can be observable in Fig. 3d and g in the main text. These non-periodic chain structures are uncontrollable in their sequences, and can convert into the alternating copolymers via stoichiometric control by post-addition of the monomers. Given that the main concern of this work lies in the selective preparation of the sequence-controlled organometallic copolymer by stoichiometric control, the formation of the less ordered organometallic chains is beyond our scope in this manuscript, and hence is not discussed in detail. We have added one sentence describing the less ordered chain structures in the main text (Paragraph 1, Page 10) as below:

“The co-existent homo- and co-polymeric segments might randomly appear in the same chain to form less ordered chain structures.”

5. *The effect of the Ag adatoms in the copolymerization should be discussed in a bit more detail. Does its activity affect the stoichiometry ratio?*

Author reply:

We thank the Reviewer for the helpful comment. Following the Reviewer’s suggestion, we have added a paragraph in the main text (Paragraph 1, Page 20) to discuss the effect of the Ag adatoms:

“The crucial role played by the Ag adatoms in the sequence-controlled copolymerization is twofold. On the one hand, the reversibility of the reactions between the Ag adatoms and the A*/P* residues enables the thermodynamic control of the final products. As a thermodynamic strategy, the change in the stoichiometric ratio of the monomers can efficiently tune the polymerization selectivity. On the other hand, the difference in the formation energies of the Ag-involved organometallic species, i.e., A-Ag-A, P-Ag-P and A-Ag-P, determines the polymerization equilibrium and hence the final organometallic product. This leads to the preferential generation of the copolymeric product once both reactive A* and P* are available on the surface ($r > 1$) because the formation of the A-Ag-P node is more energetically favorable.”

The activity of the Ag adatom does affect its reaction rate (i.e., kinetics) with different monomers, but should not change the final product and the corresponding stoichiometric ratio which are thermodynamically determined according to our mechanistic analyses.

6. *How general is the results on similar systems? The ratio is an integer number 2, is it accidental or valid for similar systems, such as other metal surfaces? And what is its limitation, is this ratio depending on other parameters such as annealing temperature?*

Author reply:

Thanks for the helpful comment that includes several concerns. In the following, we’d like to address them one by one.

(1) The universality of the stoichiometric strategy

To explore whether the stoichiometric strategy can take effect in other similar surface-confined

polymerization systems, we have carried out additional experiments using either a different metal substrate or varied aromatic bromide and terminal alkyne monomers. The results show a similar reaction scenario in the Ag(111)-supported system composed of different monomers, but distinct reaction pathways on a different substrate.

First of all, we have explored the reaction between A-H and P-Br on a different substrate, Au(111), and found that the reaction process and products on Au(111) are distinct from those on Ag(111). No significant changes in the P-Br and A-H monomers (Fig. R2a) are observed after annealing of the Au(111) sample at mild temperatures (e.g., ~ 320 K). It's been demonstrated in previous studies (*J. Am. Chem. Soc.* **2019**, *141*, 4824; *J. Phys. Condens. Matter* **2016**, *28*, 083002) that the debromination of the aromatic bromides on Au(111) manifests a higher energy barrier than that on Ag(111). Therefore, the debromination of P-Br is impeded at mild temperatures on Au(111), and the deprotonation of A-H via its reaction with the debrominated P* residue, as demonstrated on Ag(111) in this work, is hence not accessible. As a result, no chemical conversion of the monomers is observed under mild temperatures. Only if the sample is thermally treated at > 400 K can the irreversible and uncontrolled intermolecular covalent reactions between A-H and P-Br on Au(111) be triggered, leading to disordered covalent products (Fig. R2b).

Fig. R2 STM images of the Au(111) surface (a) after co-evaporation of A-H and P-Br ($V = 100$ mV and $I = 100$ pA), and (b) after annealing of the sample at ~ 410 K ($V = 100$ mV and $I = 100$ pA). Insets of (a): High-resolution STM images of the P-Br (upper panel, $V = 10$ mV and $I = 100$ pA) and A-H (lower panel, $V = 100$ mV and $I = 100$ pA) monomers. Scale bars: (a) 10 nm, (b) 8 nm.

Subsequently, we have studied the reaction between different aromatic bromide and terminal alkyne monomers on Ag(111). The results reveal a similar reaction scenario in the new system as that demonstrated between A-H and P-Br in the main text. For the details, please refer to the “Additional Experimental Results” section above on Page R3. The corresponding revisions can be found on Page 20 in the main text and Pages S9-S11 in the Supplementary Materials.

However, it should be pointed out that the confirmation of the universality of the stoichiometric strategy requires exhaustive case studies of different surface-confined systems. Obviously, such studies cannot be completed at the current stage. Therefore, we'd like to leave this open for further explorations in the future.

(2) The origin of the specific stoichiometric ratio, $r = 2$

It's not a coincidence that the yield of the alternating copolymer maximizes at an integer ratio of $r = 2$. In fact, the specific stoichiometric ratio is a result of the reaction mechanism between A-H

and P-Br on Ag(111), as discussed in detail in the main text (Pages 16-18) and Supplementary Materials (Page S8). It's demonstrated that once P* involved in the reaction is twice as A* (i.e., $r = 2$), a half of the P* residues are consumed by the activation of the A-H monomer and the other half preferentially copolymerize with the activated A*, giving rise to the selective formation of the alternating $[A-Ag-P-Ag]_n$ product. Such an ideal model is confirmed by our experimental finding that $[A-Ag-P-Ag]_n$ becomes the dominant product at $r \sim 2$. Note that there are inevitable errors for the experimentally achieved stoichiometric ratios, and hence the tilde symbols “ \sim ” are used in front of the experimentally achieved r values in the manuscript.

(3) The limitation of the stoichiometric strategy

As a thermodynamic approach to steer the surface-confined reaction selectivity, the stoichiometric strategy may fail in kinetically controlled reactions. The validity of the stoichiometric strategy may rely on other reaction parameters like the annealing temperature. For the reaction between A-H and P-Br on Ag(111), their organometallic polymerization under mild temperatures ranged from RT to 340 K is thermodynamically controlled and shows an r dependence. In contrast, the kinetically controlled irreversible covalent reactions between P-Br and A-H take place at elevated temperatures (e.g., ~ 410 K), leading to disordered covalent products (Fig. R3), and the stoichiometric strategy is therefore not applicable.

Fig. R3 STM image of the less ordered covalent products obtained by thermal annealing of the A-H + P-Br sample at ~ 410 K ($V = 100$ mV and $I = 100$ pA). Scale bar: 15 nm.

Fig. R3 has been added to the Supplementary Materials as Supplementary Fig. 2. A sentence referring to the figure is added to the main text (Paragraph 1, Page 10) as below:

“Moreover, further annealing of the A-H + P-Br samples at elevated temperatures (e.g., ~ 410 K) led to less ordered covalent structures (Supplementary Fig. 2).”

7. *What are the coverage ratios per unit area for the different types of polymerization cases? What are the effects of the inter-chain coupling in the process?*

Author reply:

The number of the molecular moieties per unit area (i.e., the coverage ratio mentioned by the Reviewer) of the close-packed assemblies formed by different organometallic polymers are 0.46 nm^{-2}

² for the A* moieties in the [A-Ag]_n assembly, 0.30 nm⁻² for both A* and P* in the [A-Ag-P-Ag]_n assembly, and 0.62 nm⁻² for P* in the [P-Ag]_n assembly.

We have two understandings for the “inter-chain coupling” mentioned by the Reviewer, that is, (1) the head-to-head organometallic reaction between two chains, and (2) the side-by-side weak interaction between the parallelly aligned chains.

In the first case, the direct coupling of two molecular chains is unlikely to happen on Ag(111), as the diffusion energy barrier for a molecular chain is usually too high to be overcome at the mild reaction temperature employed in this work. Instead, it's more likely that the growth and transformation of the polymeric chains take place via a recrystallization-like process involving the bilateral dissociation and recombination of the organometallic nodes and the diffusion of the dissociated monomeric molecular moieties.

In the second case, the inter-chain interaction between the parallelly aligned chains seems to have an impact on the reaction process. For example, the [P-Ag]_n + A-H and [A-Ag]_n + P-Br reactions are prone to occur at the endings of the chains, as observed in Fig. R1b and c. The reason may stem from an easier approach of the monomers and Ag adatoms to the reactive sites at the endings of the polymeric chains than to those in the middle where a steric hinderance may originate from the side-by-side inter-chain interaction. According to our reply to Comment 6, the inter-chain interaction may also contribute to the stabilization of the polymeric products.

8. Can the authors provide some large-scale topographic images to get better views of the surface homogeneity?

Author reply:

Fig. R4 given below presents representative large-scale STM images of the samples at $r \sim 1$, 2 and 6.5.

Fig. R4 Large-scale STM images of the reaction products achieved by P-Br and A-H on Ag(111) at (a) $r \sim 1$ ($V = 10$ mV and $I = 100$ pA), (b) $r \sim 2$ ($V = 100$ mV and $I = 70$ pA), and (c) $r \sim 6.5$ ($V = 100$ mV and $I = 100$ pA). The assemblies of the by-product, P-H, and the [A-Ag-P-Ag]_n segments are marked by the dashed black and light blue frames in (b) and (c), respectively. Scale bars: 16 nm.

Reviewer #2

The manuscript by L. Xing and co-workers reports on a very interesting case of on-surface

polymerization and stoichiometric control, as studied by scanning tunneling microscopy and DFT calculations. The monomers 4,4''-dibromo-p-terphenyl (P-Br) and 2,5-diethynyl-1,4-bis(phenylethynyl)benzene (A-H) with Ag adatoms on Ag(111) surface form the phenyl-silver-phenyl (P-Ag-P) and alkynyl-silver-alkynyl (A-Ag-A) intermolecular organometallic connections via the dehalogenated and dehydrogenated metalation reactions. The manuscript is well written, and the data presented are of high quality. Overall, the manuscript is interesting but would require minor revisions and I have only three concerns in relation to this study.

Author reply:

We thank the Reviewer for the overall positive and suggestive comments of our manuscript. Actually, we find two rather than three comments raised by the Reviewer.

- 1. The main point of this work is stoichiometric control of the polymerization by stoichiometric dosage with high selectivity. Does the stoichiometric control only work for the specific precursor? Is it applicable for different on-surface systems with different monomers? If possible experimental evidence on the universality by using slightly modified monomer should be provided to get further insight.*

Author reply:

We appreciate the Reviewer's insightful comment. Following the Reviewer's suggestion, we have carried out additional experiments using varied aromatic bromide and terminal alkyne monomers. The results reveal a similar reaction scenario in the new system as that demonstrated between A-H and P-Br in the main text. For the details, please refer to the "Additional Experimental Results" section above on Page R3. The corresponding revisions can be found on Page 20 in the main text and Pages S9-S11 in the Supplementary Materials.

However, it should be pointed out that the confirmation of the universality of the stoichiometric strategy requires exhaustive case studies of different surface-confined systems. Obviously, such studies cannot be completed at the current stage. Therefore, we'd like to leave this open for further explorations in the future.

- 2. In general, on-surface synthesis sensitively depends on the temperature of substrate. why not author studied on surface synthesis by varying temperature? Also, I have concern about simulation that What is the temperature in simulation, room temperature?*

Author reply:

Thanks for the comment. According to the Reviewer's suggestion, we have changed the annealing temperatures to show that the stoichiometric ratio controlled organometallic polymerization can take place at mild temperatures ranged from RT to 340 K. We have now added one sentence to the "Methods" section on Page 21 to clarify the reaction temperature as below:

"The surface-confined organometallic reactions can be initiated by thermal treatments of the samples at mild temperatures ranged from RT to 340 K."

We have also carried out the surface-confined polymerization at higher temperatures, e.g., 410 K. The result is added to the main text and Supplementary Materials as below:

Paragraph 1, Page 10 in the main text: “Moreover, further annealing of the A-H + P-Br samples at elevated temperatures (e.g., ~ 410 K) led to less ordered covalent structures (Supplementary Fig. 2).”

Supplementary Fig. 2 STM image of the less ordered covalent products obtained by thermal annealing of the A-H + P-Br sample at ~ 410 K ($V = 100$ mV and $I = 100$ pA). Scale bar: 15 nm.

The DFT simulation of the surface-confined reaction was conducted at 0 K in this work, which is quite routine for the theoretical computations of the surface-confined reactions (*J. Phys. Condens. Matter* **2016**, 28, 083002). It's been demonstrated previously (*J. Phys. Chem. C* **2016**, 120, 21716) that there is no obvious temperature effect on the theoretical simulation of the reaction pathway for a surface-confined reaction without any products desorbed, which is exactly the case in this work. Therefore, we are convinced that the DFT results at 0 K should be reliable for understanding the reaction mechanism in this work.

Reviewer #3

Xing and coauthors reported the selective alternating copolymerization between an aromatic halide and a terminal alkyne on Ag(111) by combined STM and DFT studies. The authors explored the surface organometallic polymerization selectivity by the stoichiometric control of two components and claimed that selectivity preference of hetero-intermolecular organometallic connection is achieved at stoichiometric ratio of 2. The thermodynamic control of the homo-/co-polymerization is realized by tuning of the stoichiometric ratio both experimentally and theoretically. Totally this is an interesting study which certainly contributes to the emerging "on-surface chemistry" field. The idea is original and the manuscript is well written. However, several major concerns as listed below prevent me from recommending its publication in Nature Communications at this stage.

- 1. The background topic is the sequence-controlled copolymerization, which is of importance in synthetic chemistry. Within the field of on-surface chemistry, this work deals with the selective alternating organometallic copolymerization via intermolecular metalation of two components with Ag atoms on Ag(111) surfaces. The lack of discussion on the potential application/transferring from controlled organometallic reactions to real covalent polymerization significantly reduces its novelty and applicability.*

Author reply:

We appreciate the Reviewer's comment. To explore the conversion from the organometallic polymers to the covalent structures, we have thermally treated the Ag(111) substrate covered by the organometallic polymers formed by A-H and P-Br at ~ 410 K. The result is added to the main text and Supplementary Materials as below:

Paragraph 1, Page 10 in the main text: "Moreover, further annealing of the A-H + P-Br samples at elevated temperatures (e.g., ~ 410 K) led to less ordered covalent structures (Supplementary Fig. 2)."

Supplementary Fig. 2 STM image of the less ordered covalent products obtained by thermal annealing of the A-H + P-Br sample at ~ 410 K ($V = 100$ mV and $I = 100$ pA). Scale bar: 15 nm.

Although the ordered covalent polymeric structures are not achieved in this work, the realization of the sequence-controlled organometallic polymerization registers its own significance. The introduction of metal centers into the covalent polymers enables not only the improved mechanical and electronic properties of the latter, but also new functionalities such as bioactivity, magnetism, conductivity, non-linear optics, ferroelectrics, catalysis, energy storage and sensitization as well (*Chem. Soc. Rev.* **2016**, *45*, 5311). Similar to covalent polymer systems, the structure and sequence of the organometallic polymeric materials dictate their properties and performance. Therefore, the exploration for sequence-controlled organometallic polymerization is of great importance. To stress the significance of the sequence-controlled organometallic polymerization, we have now added a sentence to the main text (Paragraph 1, Page 21) as below:

"The sequence-controlled organometallic polymerization may not only improve the mechanical and electronic properties of the organic polymers, but also achieve promising functionalities such as conductivity, non-linear optics, catalysis and so on and so forth⁴⁷."

2. To clarify the specific P-Br-related species that takes effect on promoting the C-H bond activation in A-H, the authors performed a series of control experiments. With regard to the reaction of A-H with $[P-Ag]_n$, in Fig. 4b and 4d, STM data verified the efficient activation of the alkynyl C-H bond in A-H at RT. In the later discussion, authors attributed it to P^* which deviates with the dissociation of P-Ag species. However, based on the reaction temperature (400 K) and calculated energy barrier (0.88 eV) of C-Ag dissociation in the cited paper (*Angew. Chem. Int. Ed.* *56*, 12852-

12856 (2017)), P^* is not available in the reaction system of A-H with $[P-Ag]_n$ at RT. Please comment on this concern.

Author reply:

We thank the Reviewer for the comment. The P^* formation in the $[P-Ag]_n + A-H$ reaction system is experimentally confirmed by the observation of the three-lobed molecular structures at the early reaction stage, as highlighted by the dashed white circles in Fig. R5. Similar structures were reported in the literature mentioned by the Reviewer (*Angew. Chem. Int. Ed.* **2017**, 56, 12852) and were identified as the trimers of the debrominated phenyl residues (P^*) which was generated by the dissociation of the P-Ag-P species and anchored to the substrate. Figure R5 can be found as Supplementary Fig. 5 in the Supplementary Materials. We have revised the related statement in the main text (Paragraph 1, Page 15) as below:

“Therefore, it’s rational to propose that P^* be the effective species promoting the C-H bond activation of A-H at RT because P^* is available in both reaction systems that comprise the P-Br monomers (via debromination^{23,41}) and $[P-Ag]_n$ chains [via dissociation of the organometallic species³⁹, as experimentally confirmed by the observation of P^* at the early reaction stage (Supplementary Fig. 5)], ...”

Fig. R5 STM image of the $[P-Ag]_n + A-H$ sample at the early reaction stage ($V = 100$ mV and $I = 100$ pA). Inset: High-resolution STM image of a phenyl residue trimer ($V = 10$ mV and $I = 100$ pA). Scale bar: 12 nm.

The P^* formed by $[P-Ag]_n$ dissociation at RT can be understood by the transition state theory where the reaction rate dramatically decays as the temperature decreases. Therefore, a reaction with a relatively high energy barrier is decelerated rather than completely inhibited at low temperatures. This conclusion is consistent with our experimental finding that the conversion of the P-Ag-P species at RT in this work (usually > 10 hours for complete reactions) is much slower than that at 400 K (5 min) reported in the literature. At a low reaction rate, the dissociation equilibrium of $[P-Ag]_n$ in the system studied in this work,

can be shifted rightward by the irreversible consumption of P^* due to its reaction with A-H,

as demonstrated in the main text. As a result, the A-H monomers can be efficiently activated by their

reaction with the $[P-Ag]_n$ chains, during which the P^* residues generated from the slow dissociation of the P-Ag-P species at RT should play a key role. Accordingly, we have now added one sentence to the main text (Paragraph 1, Page 16) as below:

“Such an irreversible consumption of the P^* residues would promote their formation by shifting the $[P-Ag]_n$ dissociation equilibrium in the $[P-Ag]_n + A-H$ reaction system.”

3. *In the discussion part of competition between homo-polymerization and co-polymerization, the reaction mechanism is explained in detail by comparing reaction energies, rationalizing the selective formation of the alternating $[A-Ag-P-Ag]_n$ product at $r = 2$. But when $r > 2$, it seems that for A-H there will always be 100% formation yield of $[A-Ag-P-Ag]_n$, although the overall yield will decrease as excess P-Br increases. Is that obvious, or does the increase of P-Br have other side effects when $r > 2$.*

Author reply:

We thank the Reviewer for the comment. The 100% transformation of A-H to $[A-Ag-P-Ag]_n$ at $r > 2$ refers to that ideally, all A-H monomers are engaged in the energetically preferred A-Ag-P species. In real reactions, however, the transformation ratio from A-H to A-Ag-P may slightly deviate from the ideal value, 100%, due to some kinetic factors (e.g., assembly-impeded molecular diffusion). The formed A-Ag-P nodes may appear in either the alternating oligomers or extended $[A-Ag-P-Ag]_n$ chains, each of which may randomly connect to the $[P-Ag]_n$ chains, as shown in Fig. 3h in the main text. We have therefore revised the sentence in the main text (Paragraph 1, Page 9) to clarify the structure of the A-Ag-P species when $r > 2$ as below:

“In addition to the main product $[P-Ag]_n$, the A-Ag-P oligomers and $[A-Ag-P-Ag]_n$ segments, as shaded in light blue in Fig. 3h, are also observed to randomly embed in the $[P-Ag]_n$ chains.”

The large amount of P-Br at $r > 2$ would facilitate the copolymerization due to a more feasible approach of the activated A^* residues to P^* . We have now added one sentence to the main text (Paragraph 1, Page 18) to clarify the possible effect of the excessive P-Br as below:

“At $r > 2$, the large amount of P-Br would facilitate the formation of the copolymeric product due to a more feasible approach of the activated A^* residues to the P^* ones. The excessive P^* species that are not engaged in the copolymerization then undergo homopolymerization to form the $[P-Ag]_n$ product.”

4. *The universality of the stoichiometric control as a synthetic strategy is not confirmed. Does the findings only work for the specific (halide and alkyne) system? Prospect on the universality of the strategy can be addressed.*

Author reply:

We appreciate the Reviewer's comment. To explore whether the stoichiometric strategy can take effect in other similar surface-confined polymerization systems, we have carried out additional experiments using varied aromatic bromide and terminal alkyne monomers. The results reveal a similar reaction scenario in the new system as that demonstrated between A-H and P-Br in the main text. For the details, please refer to the “Additional Experimental Results” section above on Page R3. The corresponding revisions can be found on Page 20 in the main text and Pages S9-S11 in the

Supplementary Materials.

However, it should be pointed out that the confirmation of the universality of the stoichiometric strategy requires exhaustive case studies of different surface-confined systems. Obviously, such studies cannot be completed at the current stage. Prospectively, we expect that as a thermodynamic approach to tune the surface reaction selectivity, the stoichiometric ratio may also play a role in other thermodynamically controlled surface systems. Nevertheless, no firm conclusion can be reached before further investigations, which we'd like to leave for the future.

5. *In addition, there are three minor issues the authors should address:*

5-a. *Since the stoichiometric ratio between two monomers is the main parameter to steer the surface organometallic polymerization selectivity, the authors should describe the precise way to determine the ratio. Or it is roughly counted from STM images?*

Author reply:

According to the Reviewer's suggestion, we have now added a paragraph in the Supplementary Materials (Paragraph 1, Page S8) to describe how we determine the stoichiometric ratios as below:

“1. Determination of Stoichiometric Ratios

The stoichiometric ratio between the P-Br and A-H monomers was experimentally controlled by their evaporation times at specific evaporation fluxes, and then confirmed by exhausting counting of both monomer numbers on the sampled surface. The STM images used for the statistical analyses are acquired at random spots of the substrate surface. The ratio value, r , is calculated by the weighted average of the P-Br molecular density in the assembled and disordered structures on the surface divided by that of A-H. The calculated molecular density of an ordered molecular assembly is the number of the molecules within a unit cell divided by the unit cell area. The molecular density of a less ordered zone is estimated by the total number of the molecules divided by the zone area that is experimentally monitored.”

The phrases referring to the statement are now added to the main text (Paragraph 2, Page 6):

“The attempt aiming at selective alternating copolymerization began with the reaction between the P-Br and A-H monomers at a ratio of $r \sim 1$ (details for r determination are provided in the Supplementary Discussion) on Ag(111) ...”

5-b. *What is the temperature of substrates when initially co-depositing molecules on surfaces, for example, in Fig. 2. And what is the RT annealing time to initiate the surface-confined organometallic (co)polymerization.*

Author reply:

The temperature of the substrate was held below RT during evaporations of P-Br and A-H. The statement can be found in the “Methods” section in the main text. We have also added one sentence in the “Methods” section on Page 21 in the main text to describe the RT annealing time:

“Specifically, the annealing at RT was usually conducted for more than 10 hours to ensure the establishment of the reaction equilibrium.”

5-c. How do authors calculate the experimental yields in Fig. 6c.

Author reply:

We have described how we determine the experimental yields of different polymeric products on Page S9 in the Supplementary Materials as below:

“The yields of the $[A-Ag]_n$ ($Y_{[A-Ag]_n}$), $[A-Ag-P-Ag]_n$ ($Y_{[A-Ag-P-Ag]_n}$) and $[P-Ag]_n$ ($Y_{[P-Ag]_n}$) products are defined as the proportions of the intermolecular A-Ag-A, A-Ag-P and P-Ag-P connections to the total amount of the organometallic nodes in the system, correspondingly. To estimate the product yields of a specific sample, the STM images at random locations of the sample are first acquired (total scanning area $\geq 32000 \text{ nm}^2$). Then, the molecular densities of the A-Ag-A (d_{A-Ag-A}), A-Ag-P (d_{A-Ag-P}), and P-Ag-P (d_{P-Ag-P}) nodes in assembled or less ordered structures are estimated. The molecular density in the ordered molecular assemblies is calculated by dividing the number of the organometallic nodes in a unit cell by the area of the unit cell. The molecular density in the less-ordered regions is estimated by dividing the total number of the organometallic nodes by the total area of the less ordered regions that are experimentally inspected ($\geq 2000 \text{ nm}^2$). The coverage proportion (C) of each ordered or less ordered structure is estimated by dividing the area of the specific structure by the total area that is experimentally inspected. The overall molecular densities of the organometallic nodes (D_{A-Ag-A} , D_{A-Ag-P} and D_{P-Ag-P}) are correspondingly estimated as:

$$D_{A-Ag-A} = \sum_i C_i d_{A-Ag-A_i},$$

$$D_{A-Ag-P} = \sum_i C_i d_{A-Ag-P_i},$$

$$\text{and } D_{P-Ag-P} = \sum_i C_i d_{P-Ag-P_i}.$$

$Y_{[A-Ag]_n}$, $Y_{[A-Ag-P-Ag]_n}$ and $Y_{[P-Ag]_n}$ are therefore calculated as:

$$Y_{[A-Ag]_n} = D_{A-Ag-A} / (D_{A-Ag-A} + D_{A-Ag-P} + D_{P-Ag-P}),$$

$$Y_{[A-Ag-P-Ag]_n} = D_{A-Ag-P} / (D_{A-Ag-A} + D_{A-Ag-P} + D_{P-Ag-P}),$$

$$\text{and } Y_{[P-Ag]_n} = D_{P-Ag-P} / (D_{A-Ag-A} + D_{A-Ag-P} + D_{P-Ag-P}).”$$

3. Other Non-Scientific Revisions

In addition to the above-mentioned revisions, we have also made several non-scientific changes to the main text and Supplementary Materials, as listed below:

Abstract, Page 2

- The phrase “in a surface-confined system” is replaced by “**on surface**”.
- The sentence “The microscopic characterizations rationalize the mechanism of the stoichiometry-dependent polymerization selectivity, providing a delicate explanation of the experimental results.” is replaced by “The microscopic characterizations rationalize the mechanism, providing a delicate explanation of the **stoichiometry-dependent polymerization**.”

Subheading, Page 6

- The words “P-Br and A-H” in the subheading are deleted.

Subheading, Page 10

- The subheading is revised as “**Homo- to Co-polymerization Transformation by Monomer Post-Addition**” to comply with the character limit.

Paragraph 1, Page 15

- The phrases “**an activation energy of**” and “**a reaction energy of**” are added.

Page 19

- The subtitle “Conclusion” is replaced by “**Discussion**”.

Paragraph 2, Page 21

- The sentences “The involvement of the intermolecular organometallic bonding introduced reversibility into the reaction system under mild conditions. Such a reversibility allowed the thermodynamic control of the homo-/co-polymerization selectivity, as realized by tuning of the stoichiometric ratio, which resulted in the highly selective alternating copolymerization at $r = 2$.” are replaced by “**As a result**, highly selective alternating copolymerization **was achieved** at $r = 2$.”, as a paragraph is added to somewhere else to discuss the role of the reaction reversibility in detail.

Paragraph 1, Page S9 in the Supplementary Materials

- The phrase “**as a result**” is added.

Reviewers' Comments:

Reviewer #1:

Remarks to the Author:

The authors have addressed my comments and the information is helpful to align their experiment data and corresponding explanation. I am satisfied with the manuscript.

Reviewer #2:

Remarks to the Author:

The manuscript was improved and recommended for publication right now.

Reviewer #3:

Remarks to the Author:

My questions are properly addressed. I recommend for the acceptance.